



# Bioturbation has a limited effect on phosphorus burial in salt marsh sediments

Sebastiaan J. van de Velde[1,2], Rebecca K. James[3], Ine Callebaut[4], Silvia Hidalgo-Martinez[5], and Filip J.R. Meysman[5,6]

[1]Bgeosys, Geoscience, Environment & Society, Université Libre de Bruxelles, Brussels, Belgium
[2]Operational Directorate Natural Environment, Royal Belgian Institute of Natural Sciences, Brussels, Belgium
[3]Groningen Institute for Evolutionary Life Sciences, University of Groningen, Groningen, The Netherlands
[4]Analytical, Environmental and Geo-Chemistry, Vrije Universiteit Brussel, Brussels, Belgium
[5]Department of Biology, Universiteit Antwerpen, 2610 Wilrijk, Belgium
[6]Department of Biotechnology, Technical University of Delft, Delft, The Netherlands

**Correspondence:** svandevelde@naturalsciences.be,F.J.R.Meysman@tudelft.nl

**Abstract.** It has been hypothesised that the evolution of animals during the Ediacaran-Cambrian transition had a major impact on atmospheric $O_2$ and $CO_2$ concentrations. The models upon which this hypothesis rests, critically assume that bioturbation by the newly evolved fauna increased the burial of organic phosphorus ($P_{org}$) within the seafloor, relative to organic carbon ($C_{org}$) and that inorganic phosphorus ($P_{inorg}$) burial was not affected by bioturbation. This assumption is centrally based on

5 data compilations from marine sediments deposited under oxic and anoxic bottom waters. Since anoxia excludes the presence of infauna and sediment reworking, the observed differences in P burial are assumed to be solely driven by the presence of bioturbators. This reasoning however ignores the potentially confounding impact of bottom water oxygenation on phosphorus burial. Here, our goal is to provide a field verification for the idea that bioturbation increases the relative burial of organic phosphorus, while accounting for bottom water oxygenation. We present solid-phase phosphorus speciation data from salt

10 marsh ponds with and without bioturbation (Blakeney salt marsh, Norfolk, UK). In both cases, the pond sediments are exposed to oxygenated bottom waters and so the only difference is the presence/absence of bioturbating macrofauna. Our data reveal that both the $C_{org} : P_{org}$ ratio of buried organic matter and the rate of $P_{inorg}$ burial are indistinguishable between bioturbated and non-bioturbated sediments. The absence of a clear effect of bioturbation on total P burial implies that previous studies may have overestimated the impact of the rise of bioturbation on atmospheric $O_2$ and $CO_2$ concentrations in the early Cambrian.

## 1  Introduction

The evolution of animals near the Ediacaran-Cambrian transition ($\sim$542 million years ago - Ma) was a major evolutionary event in Earth's history (Mangano and Buatois, 2017; Meysman et al., 2006; Wood et al., 2019). Early benthic animals developed the





ability to burrow (the so-called 'burrowing revolution'; Meysman et al., 2006), which profoundly changed the geochemical cycling and burial of elements in the seafloor (Aller, 1977; McIlroy and Logan, 1999; Meysman et al., 2006).

Burial of carbon (C) and phosphorus (P) within marine sediments plays a key role in the long-term functioning of the Earth system through regulation of atmospheric $O_2$ and $CO_2$ concentrations (Bergman et al., 2004; Berner, 1982). Therefore, an

important open question is to what extent the burrowing revolution may have affected marine $C$ and $P$ burial, and whether this impact was substantial enough to influence the atmospheric composition and climate in the early Cambrian. In recent modelling studies, it has been hypothesised that the rise of bioturbation may have increased atmospheric $CO_2$ concentrations and decreased atmospheric $O_2$ concentrations, thus inducing warmer climatic conditions and more widespread ocean anoxia (Boyle et al., 2014; van de Velde et al., 2018). These modelled effects are, however, critically dependent on the way that the

effect of bioturbation on phosphorus burial in coastal and shelf sediments is described (Boyle et al., 2014; van de Velde et al., 2018). The key assumption is that bioturbation decreases the carbon-to-phosphorus ratio of the organic matter ($C_{org} : P_{org}$) that is buried in marine sediments. The additional removal of P then decreases productivity in the ocean (which is P-limited), thereby limiting the overall burial of organic carbon in the seafloor, which subsequently increases atmospheric $CO_2$ and lowers atmospheric $O_2$ (Boyle et al., 2014; van de Velde et al., 2018). A second (and implicit) assumption is that bioturbation does

not directly affect the burial of inorganic phosphorus ($P_{inorg}$). Instead, $P_{inorg}$ burial is assumed to only depend on the amount of organic matter that rains down on the sediment (Boyle et al., 2014).

Neither of these assumptions have been extensively verified under field or laboratory conditions. The first assumption (bioturbation lowers the $C_{org} : P_{org}$ of buried organic matter) is based on a data compilation from present-day sediments deposited under oxic versus anoxic water columns. Sediments deposited under an oxygenated water column typically have a lower

$C_{org} : P_{org}$ (range: 30-250) than sediments deposited under an anoxic water column (range: 300-700) (Slomp and Van Cappellen, 2007). Within the present-day seafloor, sediments deposited under oxic conditions generally experience bioturbation (Levin et al., 1991), and so it is argued that the observed differences in $C_{org} : P_{org}$ are primarily the consequence of bioturbation, and less due to differences in the redox state (i.e. the oxygenation) of the overlying water (Boyle et al., 2014; Dale et al., 2016). The second assumption (bioturbation does not affect the burial of inorganic phosphorus) is based on the argument

that the burial of inorganic phosphorus in the form of minerals such as apatite ($Ca_5(PO_4)_3(F,Cl,OH)$) is predominantly controlled by the organic matter input into the sediment. A higher input of organic matter leads to higher mineralisation, thus inducing phosphate to be released in the pore water, which then leads to more precipitation and subsequent burial of phosphate as apatite (Ruttenberg and Berner, 1993). Nevertheless, other studies have argued that bioturbation increases the burial of authigenic apatite (Dale et al., 2016; Zhao et al., 2020) via the downward mixing of the iron oxides on which phosphate

is adsorbed (Slomp et al., 1996a). Introducing iron oxides at depth releases phosphate further away from the sediment-water interface (SWI), and provides additional adsorption sites for phosphate released from organic matter. Accordingly, phosphate is potentially retained longer in the sediment, which could stimulate the precipitation and eventual burial of inorganic phosphorus minerals (Slomp et al., 1996a).

To be able to adequately tease apart the impact of bioturbation from other confounding factors (such as the redox conditions

at the SWI or organic matter input), one requires sites that have oxygenated waters and a similar input of organic matter, but



no bioturbation. While these conditions are extremely rare in the modern seafloor, they are found in salt marsh ponds along the North Sea coast of Norfolk (UK), which contain sediments with overlying oxygenated water that are either bioturbated or non-bioturbated (Antler et al., 2019; Hutchings et al., 2019). These two different sediment types can be found in neighbouring ponds, less than five meters apart, and no systematic difference in local sediment input, organic matter supply or other boundary

conditions has been found between the two pond types (Antler et al., 2019; Hutchings et al., 2019; van de Velde et al., 2020). This remarkable biogeochemical dichotomy between the pond sediments has recently been attributed to alternative stable states, in which small initial differences between ponds are amplified through non-linear positive feedbacks in the sedimentary iron-sulphur cycle (van de Velde et al., 2020).

Whatever the cause of the biogeochemical dichotomy, the important aspect here is that by comparing the geochemistry of

the two oxygenated pond types, we can single out the effect of burrowing fauna on sediment biogeochemistry. These ponds within the Norfolk salt marsh complex, hence, provide a unique environment to study the impact of bioturbation on the burial of organic and inorganic phosphorus, without the confounding effect of bottom water oxygenation. To this end, we collected solid-phase phosphorus data during three separate visits and quantified the burial rates of P in the bioturbated and non-bioturbated ponds.

## 15  2  Materials and Methods

### 2.1  Field site

Blakeney salt marsh (Fig. 1) is part of a larger salt marsh complex along the North Sea coast of East Anglia (UK). The higher, vegetated, marsh hosts several shallow, water-filled ponds with a surface area of ∼50 - 500 $m^2$ and a water depth of 10 – 20 $cm$ (Fig. 1;  van de Velde et al., 2020). These ponds show a conspicuous dichotomy in terms of their sediment geochemistry,

and belong to either one of two end-member types. Pond sediments are either heavily bioturbated and the solid phase is rich in iron oxides, or sediments are non-bioturbated and the pore water is rich in hydrogen sulphide (Antler et al., 2019; Hutchings et al., 2019; van de Velde et al., 2020). The bioturbated ponds are colonised by large macrofauna, mostly Nereis and Arenicola, at high densities (∼1000 organisms $m^{-2}$; Antler et al., 2019), while the non-bioturbated sediments do not show signs of macrofauna or burrows. Water column concentrations of oxygen, DIC and nutrients are not statistically different between pond

types, and both types of ponds have the same sedimentation flux ($0.9 \pm 0.1$ $kg m^{-2} yr^{-1}$), indicating they receive a similar input of detrital minerals. Furthermore, all ponds display similar inputs of organic matter, thus suggesting that the only important difference between the ponds is the presence of burrowing fauna (van de Velde et al., 2020). Depth profiles of $^{137}Cs$ show a well-defined peak in the non-bioturbated ponds, suggesting they have been undisturbed by fauna for at least 60 years (van de Velde et al., 2020).

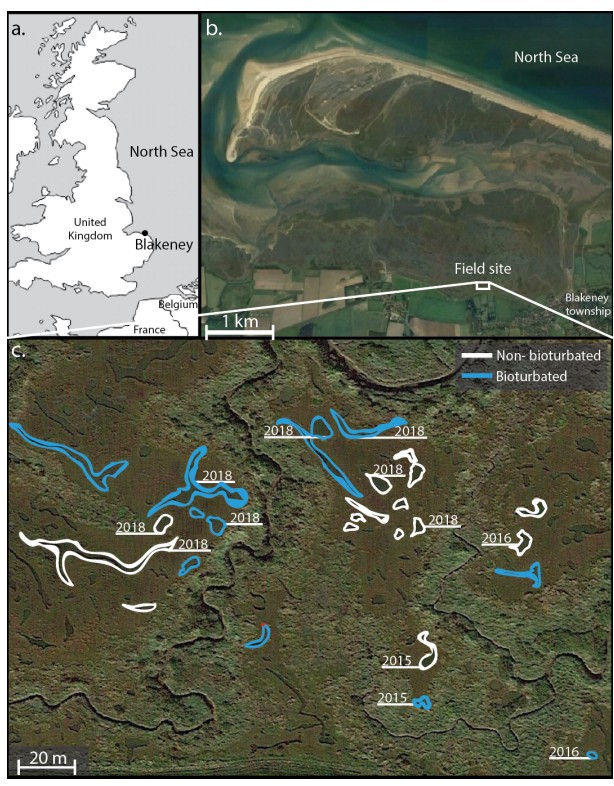

**Figure 1.** Overview of the field site in the Blakeney saltmarsh system, UK. Sampled ponds are outlined in white (non-bioturbated) and blue (bioturbated), with ponds where sediment cores taken denoted with the sampling year. Coordinates of the sampled ponds are given in Table A1. (b,c) Map data ©2020 Google.

## 2.2 Sediment sampling and analysis

Sediment cores were collected on three separate visits (October 2015, August 2016 and August 2018). Twelve ponds were examined in total (sampling sites are indicated in Fig. 1). During each sampling campaign, two replicate sediment cores were collected from each sampled pond. Core sectioning was done at 0.5 cm resolution from 0 to 3 cm depth, at 1 cm resolution

5  between 3 and 8 cm depth, and in 2 cm slices from 8 to 22 cm depth. Sediment sections were collected in 50 mL centrifuge tubes (polypropylene; TPP, Switzerland). In 2015 and 2016, sediment cores were processed under anaerobic conditions in a glove bag with $N_2$, freeze-dried and stored in a sealed aluminium bag under $N_2$ atmosphere for later solid-phase analysis. In 2018, cores were immediately sliced in open air in the field, and subsequently freeze-dried and stored under room conditions in 50 mL centrifuge tubes. This difference in sampling procedure reflects the subsequent analyses. In 2015 and 2016, solid

10  phase was analysed for different phosphorus fractions following the SEDEX phosphorus extraction (Ruttenberg, 1992; Slomp et al., 1996a). In 2018, due to the higher amount of samples, we decided to use a simpler and faster extraction method that only differentiates between inorganic and organic phosphorus fractions (Bowman, 1989; Olsen and Sommers, 1982).





Sediment samples from all three campaigns were analysed for carbon and nitrogen content. To this end, freeze-dried solid phase samples were ground to a fine powder and analysed by an Interscience Flash 2000 organic element analyser (precision <5%) for determination of particulate organic carbon (POC) and total nitrogen (TN). Before analysis, samples for POC were first acidified with 0.1M HCl to remove the inorganic carbon (Nieuwenhuize et al., 1994). Concentrations of POC and TN are

expressed as mass % of dry sediment. Organic matter $C_{org} : N_{tot}$ was calculated as the molar ratio of POC over TN. The POC and TN results have been presented previously in van de Velde et al. (2020).

The SEDEX procedure used in 2015 and 2016 separates total sedimentary P in 5 fractions; exchangeable P ($P_{exch}$), P associated with iron ($P_{Fe}$), authigenic P ($P_{auth}$), detrital P ($P_{det}$) and P associated with organic matter ($P_{org}$). All extractions were performed on a subsample of 300 mg, under room temperature and under constantly agitated conditions (the extraction

procedure is detailed in Table A2). Inorganic phosphorus ($P_{inorg}$) is calculated as

$$P_{inorg} = P_{exch} + P_{Fe} + P_{auth} + P_{det} \tag{1}$$

and expressed as $\mu$mol g$^{-1}$ dry sediment. The extraction procedure used in 2018 separates the sediment phosphorus content in 2 fractions; total phosphorus ($P_{tot}$) and inorganic phosphorus ($P_{inorg}$), organic phosphorus is then calculated as the difference between $P_{tot}$ and $P_{inorg}$ (Bowman, 1989; Olsen and Sommers, 1982). Extractions were performed on a subsample of 1 g,

under room temperature and under constantly agitated conditions (extraction procedure detailed in Table A2).

### 2.3 Burial fluxes and solid phase inventories

Burial fluxes of solid phase species were calculated based on the sedimentation flux ($J_{sed}$) and the concentration of the solid component at the bottom of the sediment column ($C_{solid}$):

$$J_{burial} = J_{sed} C_{solid} \tag{2}$$

The sedimentation flux was previously determined based on $^{210}Pb$ and $^{137}Cs$ dating, and was statistically indistinguishable between pond types (0.9 $\pm$ 0.1 kg m$^{-2}$ yr$^{-1}$; van de Velde et al., 2020).

Solid phase inventories were calculated by integrating measured concentrations over the first 20 cm of the sediment cores:

$$INV = \rho_{solidphase} \int_{x_{down}}^{x_{up}} (1 - \phi_x) C_{solid} dx \tag{3}$$

Where $\phi_x$ is the porosity at depth x and $\rho_{solidphase}$ the solid phase density (previously determined to be 2.2 g cm$^{-3}$; van de

Velde et al., 2020). The porosity depth profile was determined from the water content and solid-phase density, considering the salt content of the pore water. The water content of the sediment was determined as the difference in sediment weight before and after freeze-drying. Porosity profiles have been presented previously in van de Velde et al. (2020).

### 2.4 Statistics

The measured values of $C_{org} : N_{tot}$ and $C_{org} : P_{org}$, calculated burial fluxes of $P_{org}$ and $P_{inorg}$ and solid-phase inventories

were averaged over the duplicate cores from each pond and one-way ANOVA's were used to test for significant differences





between bioturbated and non-bioturbated ponds. Due to the different methods used for P extraction in 2015 and 2016 relative to 2018, the inclusion of the sampling year as a random effect was tested through model evaluation using AIC (Akaike information criterion). Sampling year was not significant for all dependent factors, and so it was excluded from subsequent analyses. Residuals were tested for normality and homoscedasticity, and all but $C_{org} : P_{org}$ fulfilled these assumptions. The values of

$C_{org} : P_{org}$ were log transformed before analysis.

## 3   Results

### 3.1   Bioturbation, $C_{org} : P_{org}$ values and organic phosphorus burial

Individual depth profiles of solid-phase variables (POC, TN, POP) show variation within the sediments of a given pond type, indicating spatial heterogeneity (Fig. 2). Still, averaged depth profiles of the six non-bioturbated and the six bioturbated ponds

reveal that bioturbated sediments have generally a stronger down-core gradient in POC, TN and POP (Fig. 2A-F), which is expected as sediment bio-mixing by bioturbating fauna acts to erase solid phase gradients (van de Velde and Meysman, 2016). Differences in POC, TN and POP inventories in the top 20 cm between bioturbated and non-bioturbated cores are insignificant (Fig. 2A-F; Table A4). Consistently, $C_{org} : N_{tot}$ and $C_{org} : P_{org}$ values are not significantly different between bioturbated and non-bioturbated pond types ($p > 0.1$; Table A4). Values of $C_{org} : N_{tot}$ range from 9 to 15, with an average value of 12 (2G).

These values are slightly higher than expected for sediments from fully marine settings ($C_{org} : N_{tot} < 10$; Burdige, 2006), but are consistent with sediments from temperate salt marshes, which generally have $C_{org} : N_{tot}$ values of 10 or higher (Boschker et al., 1999; Spivak et al., 2018). Similarly, $C_{org} : P_{org}$ values averaged around 500 in both the non-bioturbated and bioturbated sediment cores (2G). These $C_{org} : P_{org}$ values are higher than expected for marine sediments with similar sedimentation rates underlying oxygenated waters ($C_{org} : P_{org} \sim 200$; Slomp and Van Cappellen, 2007), and are more representative of low-

oxygen and anoxic marine environments ($C_{org} : P_{org}$ = 300-700; Slomp and Van Cappellen, 2007). However, since the pond waters were oxygenated at the time of sampling (van de Velde et al., 2020), the elevated $C_{org} : P_{org}$ values most likely reflect the contribution of plant material from the surrounding marsh ($C_{org} : P_{org} > 500$; Table A3), which is substantially elevated above the $C_{org} : P_{org}$ of marine plankton ($\sim$106; Redfield, 1934).

Overall, our organic P data show two main findings: (i) bioturbated sediments have similar amounts of $P_{org}$ ($P_{inv}$ = 0.067 ±

0.005 mmol cm$^{-2}$; Fig. 3A) compared with non-bioturbated sediments ($P_{inv}$ = 0.072 ± 0.004 mmol cm$^{-2}$; Fig. 2,3a), and (ii) $C_{org} : P_{org}$ values are not significantly lower in bioturbated sediments (relative to non-bioturbated sediments) (Fig. 2). This implies that the burial of $P_{org}$ in bioturbated sediments is comparable in bioturbated sediments, relative to non-bioturbated sediments (Fig. 3B). Indeed, the difference in $P_{org}$ burial between bioturbated (17 ± 4 $\mu$mol m$^{-2}$ d$^{-1}$) and non-bioturbated (19 ± 2 $\mu$mol m$^{-2}$ d$^{-1}$) ponds is not statistically significant ($p > 0.1$; Table A4).





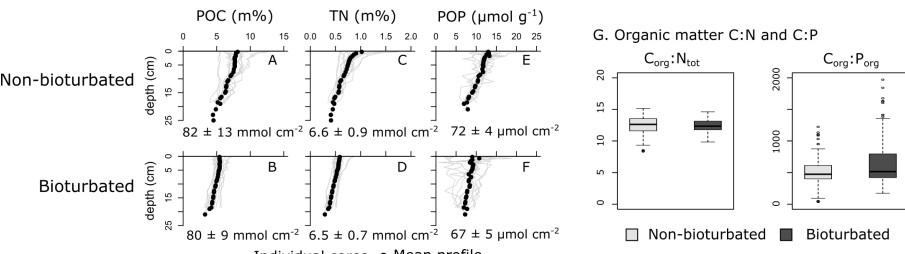

**Figure 2.** Vertical solid phase profiles of (A,B) particulate organic carbon (POC), (C,D) total nitrogen (TN) and (E,F) particulate organic phosphorus (POP). Individual cores are plotted as a light gray lines, and the average vertical profile over all cores is plotted as black dots. Values are averaged inventories and errors are 1 standard deviation. (G) Boxplots of the $C_{org} : N_{tot}$ and $C_{org} : P_{org}$ of the particulate organic matter fraction.

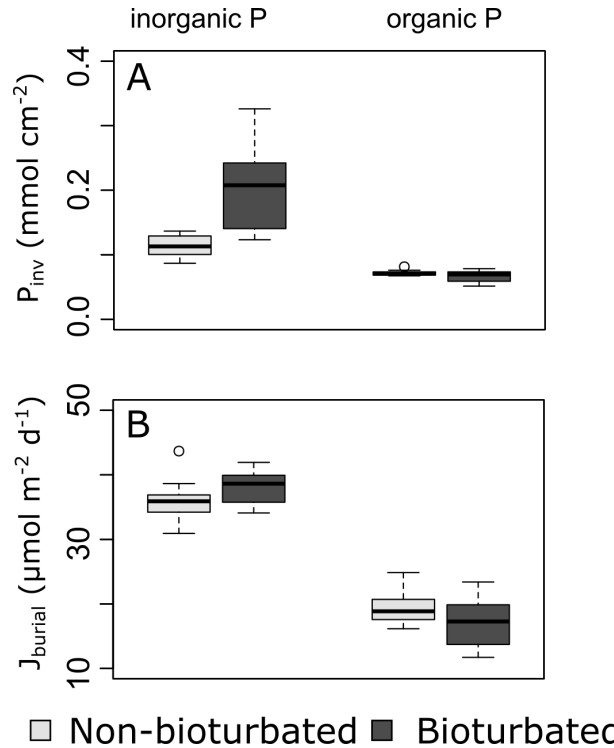

**Figure 3.** Boxplots of (A) inventories and (B) burial rates of inorganic and organic P fractions. The data of 2015 are not included in panel A, since some depth samples were not analysed for P fractionation.

### 3.2 Bioturbation and inorganic phosphorus burial

Our solid phase analyses show that bioturbated sediments ($30 \pm 10 \ \mu\text{mol g}^{-1}$) contain significantly more particulate inorganic phosphorus than non-bioturbated sediments ($16 \pm 3 \ \mu\text{mol g}^{-1}$) ($p < 0.1$; Fig. 3A; Fig. 4A,F; Table A4). SEDEX extractions,



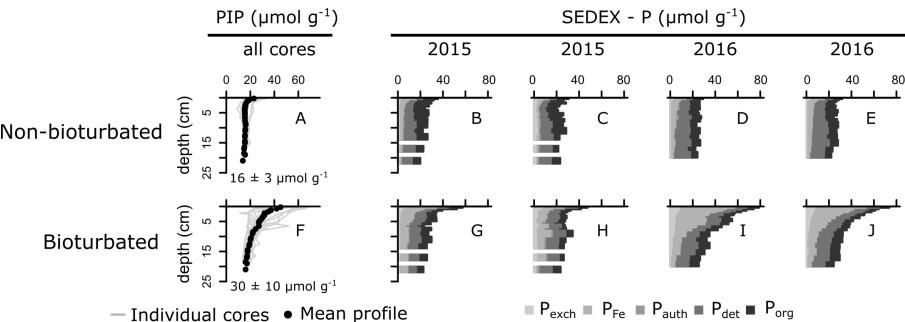

**Figure 4.** Vertical solid phase profiles of (A,F) particulate inorganic phosphorus (PIP). Values are depth-averaged concentrations. (B)-(E), (G)-(J) Solid phase phosphorus speciation (SEDEX) of cores collected in 2015 and 2016. $P_{exch}$=exchangeable phosphorus, $P_{Fe}$=iron-bound phosphorus, $P_{auth}$=authigenic phosphorus, $P_{det}$=detrital phosphorus, $P_{org}$= organic phosphorus. Note that the $P_{auth}$ fraction is too small to be visible on the figure.

performed on sediment cores collected in 2015 and 2016, show that the large difference in $P_{inorg}$ is caused by the much higher $P_{exch}$ and $P_{Fe}$ contents in the bioturbated sediments ($P_{exch} + P_{Fe}$ is up to 90 mmol P cm$^{-2}$ higher in the bioturbated sediments; Table A5). Phosphorus adsorbs onto iron oxide minerals, which are introduced at depth by the downward mixing of benthic fauna (Slomp et al., 1996a). Indeed, $\sim 50\%$ of the Fe minerals in the bioturbated sediments at Blakeney salt marsh

are in oxidised form, whereas $< 10\%$ are in the non-bioturbated sediments (see van de Velde et al. (2020) for an extended discussion of the Fe-S cycle at the field site).

The increased inventory of $P_{exch}$ and $P_{Fe}$ at depth seemingly does not lead to more precipitation of $P_{auth}$ (Table A5), since we do not find detectable formation of authigenic apatite, the concentrations of which are negligible throughout all sediment cores in both ponds ($< 1$ $\mu$mol g$^{-1}$). The lack of measurable authigenic P formations is at odds with previous hypotheses

(Dale et al., 2016; Slomp et al., 1996a). Potentially, the high concentrations of $Fe^{2+}$ in the pore water of the bioturbated sediments (up to 300 $\mu$M; van de Velde et al., 2020) promoted the formation of vivianite over apatite (Ruttenberg, 2014). Because vivianite is extracted in the $P_{Fe}$ fraction (Nembrini et al., 1983), we cannot separate between P associated with iron oxides, and P in the form of vivianite. The precipitation of vivianite in the bioturbated sediments could potentially stimulate the burial of $P_{inorg}$, but this effect is not readily seen in our data. While $P_{inorg}$ concentrations are high in the top layers, they

strongly decrease with depth, thus $P_{inorg}$ is not efficiently buried. Overall, we find that the burial rate of $P_{inorg}$ is $38 \pm 3$ $\mu$mol m$^{-2}$ d$^{-1}$ in bioturbated sediments, and $36 \pm 3$ $\mu$mol m$^{-2}$ d$^{-1}$ in the non-bioturbated sediment, and this difference is not significant ($p > 0.1$; Table A4).

## 4   Discussion

By comparing non-bioturbated and bioturbated sediment cores collected from an East Anglian salt marsh, this study provides

a field verification for two existing hypotheses: (1) bioturbation does not significantly affect burial of inorganic P minerals





and (2) bioturbation increases the burial of organic phosphorus relative to organic carbon. Related to the first hypothesis, our field data show that a bioturbated sediment contains more inorganic P, which exists mainly in the form of iron-associated P. Nevertheless, the accumulation of inorganic P occurs principally in the top layers of the sediment. At 20 cm, the $P_{inorg}$ levels in bioturbated sediments decrease to similar levels as in non-bioturbated sediments (Fig. 4). As a consequence, we find that the burial of $P_{inorg}$ was not significantly different between bioturbated and unbioturbated ponds, which does not support the idea that bioturbation stimulates the burial of inorganic P minerals. Our data are hence not congruent with previous modelling and field studies that have suggested that bioturbating fauna stimulates the burial of authigenic apatite (Slomp et al., 1996a; Zhao et al., 2020). These studies argued that bioturbators mix iron oxides, on which phosphate is adsorbed, further away from the sediment-water interface (SWI), increasing the retention time of phosphate in the sediment, and subsequently stimulating the precipitation and eventual burial of inorganic phosphorus minerals (Slomp et al., 1996a; Dale et al., 2016). Bioturbators also flush their burrows, which removes phosphate from the pore water (Dale et al., 2016). Consequently, depending on the animals community (irrigation versus mixing), the pore water can become undersaturated with respect to apatite, leading to lower precipitation rates and burial of apatite. The difference between oxic, bioturbated and oxic, non-bioturbated sediments inferred from the model study of Dale et al. (2016) is only a few $\mu$mol m$^{-2}$ d$^{-1}$, which is likely undetectable in a field study like ours (Fig. 3). Consequently, our results suggest that the effect of bioturbation on $P_{inorg}$ minerals is small, consistent with diagenetic modelling results (Dale et al., 2016).

Related to the second hypothesis, we find that the $C_{org} : P_{org}$ of buried organic matter is indistinguishable between bioturbated and non-bioturbated sediments (Fig. 2G). Accordingly, our field data do not support the idea that bioturbation increases the relative burial of organic phosphorus. It should be noted however that the $C_{org} : P_{org}$ values recorded here are relatively high, and amount to $\sim 500$ in both the non-bioturbated and bioturbated sediment cores (Fig. 2G). These $C_{org} : P_{org}$ values are higher than expected for marine sediments underlying oxygenated waters, and deviate considerably from the $C_{org} : P_{org}$ values 70-100 in oxic sediments obtained in the model study of Dale et al. (2016). One reason for this is that the sedimentation velocity in Norfolk salt marsh ponds is relatively high (0.3 cm yr$^{-1}$, compared to a mean value of 0.1 cm yr$^{-1}$ for shelf sediments; Burwicz et al., 2011). High sedimentation velocities stimulate the preservation of organic matter (Canfield, 1994), and hence cause less mineralisation of $C_{org}$, thus leading to higher $C_{org} : P_{org}$ values. But even when compared to sediments with similar sedimentation rates underlying oxygenated waters ($C_{org} : P_{org} \sim 200$; Slomp and Van Cappellen, 2007), the values remain high. The C:N:P of the organic matter in the ponds suggests a large contribution of plant material from the surrounding marsh (Fig. 2), which has elevated C:P ratios compared to marine organic matter (Table A3). Because terrestrial material is less easily degraded than marine organic matter, the addition of plant material to the sediment organic matter pool dilutes the signal of the marine organic matter (Ruttenberg, 2014). This presence of terrestrial vegetation explains the high $C_{org} : P_{org}$ values recorded here. But still, the observation remains: even in these sediments with high $C_{org} : P_{org}$ values, there is no difference in $C_{org} : P_{org}$ values between bioturbated and non-bioturbated sediments.

Low $C_{org} : P_{org}$ values observed in Phanerozoic sediments deposited under oxygenated conditions have initially been attributed to different water column redox conditions (Ingall and Jahnke, 1997; Slomp and Van Cappellen, 2007; Ruttenberg, 2014). The mechanism behind this remains still somewhat elusive, but could be related to the formation of microbial polyphos-





phates, which are generated during the breakdown of the settling organic matter under oxic conditions (Diaz et al., 2008). During diagenesis, these polyphosphate molecules would then be converted into more refractory organic P (as for example phosphate esters or phosphonates), which then constitutes a permanent $P_{org}$ sink (Berner et al., 1993; Van Cappellen and Ingall, 1994; Ingall and Jahnke, 1997). Because Phanerozoic sediments that are deposited under oxygenated conditions are vir-

tually always bioturbated, more recent studies have argued that the decreased $C_{org} : P_{org}$ values in Phanerozoic sediments are related to bioturbation, rather than water column redox conditions (Boyle et al., 2014). However, the diagenetic model results of Dale et al. (2016) show that, while $C_{org} : P_{org}$ values in oxic, bioturbated sediments are indeed much lower than in anoxic sediments ($\sim 70$ compared to $\sim 500$), the difference between oxic, bioturbated and oxic, non-bioturbated sediments is much smaller ($\sim 70$ compared to $\sim 100$). These modelled $C_{org} : P_{org}$ values suggest that the effect of bioturbation is much smaller

than the effect of water column redox, and agrees with the observations made here. The high $C_{org} : P_{org}$ values reported from anoxic sediment are hence not due a lack of bioturbation, but rather due to a redox effect, where the absence of oxygen stimulates the preservation of organic matter (increasing $C_{org}$), and reduces the formation of polyphosphates by micro-organisms (decreasing $P_{org}$).

Still, a number of caveats need to be taken into account when interpreting our results, as observations made in the surface

sediments of the Norfolk salt marsh ponds are not necessarily representative of burial signals made at several metres deep in coastal and shelf sediments. As noted above, the $C_{org} : P_{org}$ ratio of buried organic matter is high, as it is influenced by the deposition of the salt marsh vegetation with high C:P ratios, and therefore, caution is required when extrapolating our results to other coastal and shelf sediments. A difference in time scale could be another reason for the discrepancy between our field observations and the lower $C_{org} : P_{org}$ values reported elsewhere in the literature. The data reported in the literature are

collected from geological formations (see, e.g., Ingall et al., 1993) or deep drill cores (see, e.g., Slomp et al., 2004). These samples have undergone several hundreds to thousands years of diagenesis before being sampled and analysed. In contrast, based on an average sedimentation velocity of 0.3 cm yr$^{-1}$ measured via radionuclide dating (van de Velde et al., 2020), our samples represent only 60 years of sediment accumulation and they hence indicate that, after $\sim 60$ years of early diagenesis, there is no discernible difference in burial of P minerals. Diagenesis later in the burial history could still affect long-term P

burial. However, we cannot see a mechanism via which bioturbation could have an imprint on the sediment that eventually leads to differential diagenesis of P minerals later in their burial history. If bioturbation does not already create a difference within $C_{org} : P_{org}$ values in the early phase of diagenesis, it is highly unlikely that will happen afterwards. So the difference in time scale cannot explain why $C_{org} : P_{org}$ values are similar in bioturbated and non-bioturbated sediments.

Overall, more field studies and laboratory experiments are required to verify our results. Nevertheless, the absence of a

clear impact of bioturbation on P burial questions whether the modelled effects of the evolution of bioturbation in the early Cambrian, if they are based on inferred effects on C:P values of buried organic matter, do not overestimate the true impact of the 'burrowing revolution'. Our study highlights that assumptions made in Earth System models should be firmly grounded in field observations, before accurate inferences about large-scale questions can be made, such as the impact of the burrowing revolution on the composition of the atmosphere and climate.





*Data availability.* All data presented in this manuscript is available from the VLIZ data repository (doi:10.14284/419)

**Table A1.** Coordinates and type of the ponds sampled in the 2015, 2016 and 2018 field campaigns in the Blakeney salt marsh. See Fig. 1c in the main text for relative geographical location of the ponds.

|    | Coordinates | Type | Year sampled |
|----|-------------|------|--------------|
| 1  | 52° 57' 22.7"N 01° 00' 14.0"E | Bioturbated | 2015 |
| 2  | 52° 57' 23.0"N 01° 00' 14.0"E | Unbioturbated | 2015 |
| 3  | 52° 57' 22.2"N 01° 00' 16.6"E | Bioturbated | 2016 |
| 4  | 52° 57' 24.0"N 01° 00' 16.0"E | Unbioturbated | 2016 |
| 5  | 52° 57' 25.2"N 01° 00' 13.2"E | Bioturbated | 2018 |
| 6  | 52° 57' 25.3"N 01° 00' 12.5"E | Bioturbated | 2018 |
| 7  | 52° 57' 24.6"N 01° 00' 10.6"E | Bioturbated | 2018 |
| 8  | 52° 57' 24.3"N 01° 00' 10.9"E | Bioturbated | 2018 |
| 9  | 52° 57' 24.7"N 01° 00' 13.4"E | Unbioturbated | 2018 |
| 10 | 52° 57' 24.4"N 01° 00' 14.1"E | Unbioturbated | 2018 |
| 11 | 52° 57' 24.3"N 01° 00' 9.9"E | Unbioturbated | 2018 |
| 12 | 52° 57' 24.1"N 01° 00' 10.1"E | Unbioturbated | 2018 |



**Table A2.** Specifics for the sequential extractions of phosphorus. After the extraction, the sample was centrifuged (2500g for 10 min) and the supernatant was filtered (0.45 $\mu$m cellulose acetate).

| Fraction | Extraction solution | Atmosphere | Time | Analysis |
|---|---|---|---|---|
| *SEDEX Phosphorus extraction (Ruttenberg, 1992; Slomp et al., 1996a)* | | | | |
| $P_{exch}$ | 1 M $MgCl_2$ | $N_2$ | 30 min. | spectrophotometer |
| $P_{Fe}$ | 12 g sodium dithionite in 480 mL sodium acetate (1M) + 60 mL sodium bicarbonate (1 M) | $N_2$ | 8 h | ICP-OES |
| | + wash: 1 M $MgCl_2$ | $N_2$ | 30 min. | spectrophotometer |
| $P_{auth}$ | 300 mL sodium acetate (1M) + 1700 mL acetic acid (1 M) | open air | 6h | spectrophotometer |
| | + wash: 1 M $MgCl_2$ | open air | 30 min. | spectrophotometer |
| $P_{det}$ | 1 m HCl | open air | 24 h | spectrophotometer |
| | + wash: 1 M $MgCl_2$ | open air | 30 min. | spectrophotometer |
| $P_{org}$ | Combustion at 550°C | open air | 2h | |
| | 1 M $HCl$ | open air | 24 h | spectrophotometer |
| | + wash: 1 M $MgCl_2$ | open air | 30 min. | spectrophotometer |
| *One-step Phosphorus extraction (Bowman, 1989; Olsen and Sommers, 1982)* | | | | |
| $P_{inorg}$ | 1 M $H_2SO_4$ | open air | overnight | spectrophotometer |
| $P_{tot}$ | Combustion at 550°C | open air | 1h | |
| | 1 M $H_2SO_4$ | open air | overnight | spectrophotometer |





**Table A3.** Summary of salt marsh plant properties. Data was first published in van de Velde et al. (2020)

| Salt marsh properties | | | | | | | |
|---|---|---|---|---|---|---|---|
| | *Suaeda maritima* | *Salicornia radicans* | *Spartina anglica* | *Ameria maritima* | *Elytrigia atherica* | *Halimione portulacoides* | *Limonium vulgare* |
| $C_{org} : N_{tot}$ | 18 | 20 | 27 | 20 | 74 | 26 | 25 |
| $C_{org} : P_{org}$ | 682 | 492 | 669 | 554 | 1289 | 802 | 692 |

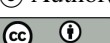



**Table A4.** Results from One-way ANOVA analysis and calculated 95% confidence intervals. Residuals were tested for normality and heterogeneity, all passed these assumptions except C:P, which was log-transformed.

| | One-way ANOVA | | | |
| --- | --- | --- | --- | --- |
| | $P_{inorg}$ burial rate | $P_{org}$ burial rate | $C_{org} : N_{tot}$ | $C_{org} : P_{org}$ |
| Pond type Mean square (df) | 13.69 (1) | 11.826 (1) | 0.04417 (1) | 0.0001 (1) |
| Residuals Mean square (df) | 6.59 (10) | 9.768 (10) | 0.22665 (10) | 0.07641 (10) |
| F-value | 2.077 | 1.211 | 0.195 | 0.0 |
| $p$ | 0.18 | 0.297 | 0.668 | 0.991 |
| **Pond type** | **95% Confidence Intervals** | | | |
| Non-bioturbated | 33.65, 38.32 | 16.38, 22.07 | 11.52, 12.39 | 6.04, 6.54 |
| Bioturbated | 35.79, 40.46 | 14.40, 20.08 | 11.40, 12.27 | 6.04, 6.54 |
| | $P_{inorg}$ inventory | $P_{org}$ inventory | POC inventory | TN inventory |
| Pond type Mean square (df) | 20006 (1) | 1.05 (1) | 28.61 (1) | 0.0033 (1) |
| Residuals Mean square (df) | 2847 (10) | 75.8 (8) | 132.14 (10) | 0.6345 (10) |
| F-value | 7.026 | 0.014 | 0.217 | 0.005 |
| $p$ | 0.029 | 0.91 | 0.652 | 0.944 |
| **Pond type** | **95% Confidence Intervals** | | | |
| Non-bioturbated | 58.26, 168.31 | 61.85, 79.81 | 72.40, 93.31 | 5.86, 7.31 |
| Bioturbated | 147.71, 257.77 | 61.20, 79.16 | 69.31, 90.22 | 5.82, 7.27 |





**Table A5.** Inventories of the individual phosphorus fractions. $P_{org}$ and $P_{inorg}$ are reported as mean ± 1 s.d., SEDEX fractions are given as a range (due to the low number of replicates) from the 2016 cores (2015 was omitted because some depth samples were not analysed for P fractionation).

| | $P_{org}$ | $P_{inorg}$ | Individual $P_{inorg}$ fractions (only 2016) | | | |
| | | | $P_{exch}$ | $P_{Fe}$ | $P_{auth}$ | $P_{det}$ |
| | $mmol\,cm^{-2}$ | $mmol\,cm^{-2}$ | $\mu mol\,cm^{-2}$ | $\mu mol\,cm^{-2}$ | $\mu mol\,cm^{-2}$ | $\mu mol\,cm^{-2}$ |
| Non-bioturbated | $0.071 \pm 0.004$ | $0.11 \pm 0.02$ | 0.13-0.19 | 30.1-32.2 | 2.4-3.8 | 100-103 |
| Bioturbated | $0.067 \pm 0.009$ | $0.20 \pm 0.07$ | 10.1-17.3 | 86.5-108 | 2.2-2.8 | 111-114 |





*Author contributions.* SJV conceived the hypothesis. SJV and FJRM organised the field sampling. All authors contributed to the field sampling. SJV and IC performed the SEDEX extractions. SJV analysed the data. RKJ did the statistical analyses. SJV and FJRM wrote the manuscript with input from all co-authors.

*Competing interests.* The authors declare that they have no conflict of interest.

5  *Acknowledgements.* This research was financially supported by the Research Foundation Flanders (FWO project grant G038819N), a TOP-BOF grant by the University of Antwerp, and the Netherlands Organization for Scientific Research (VICI grant 016.VICI.170.072 to FJRM). SJV is supported by Belgian Science Policy (FED-tWIN2019-prf-008 – ReCAP project grant). The authors would like to thank Tom Van der Spriet from the University of Antwerp for the phosphorus extraction on the 2018 sediment samples.



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
