# Peer review of "Bioturbation has a limited effect on phosphorus burial in salt marsh sediments"

_Biogeosciences, 2020_

## Referee Comment (RC1) · Anonymous Referee #1 · 8 Dec 2020

This is a well presented study that compares P burial and preservation in salt marsh ponds under oxygenated vs anoxic conditions. A nice aspect of this study is that major variables that are known to affect P deposition, such as sediment accumulation rate and organic matter composition, are very similar in the different ponds. Thus, the effect of other variables such as oxygen and bioturbation can be isolated. The paper is written in such a way that it appears a primary motivation for this work is to refute statements made in the paper by Boyle et al., 2014. From reading this paper it sounds like the Boyle paper did not consider many variables that can affect P cycling in sediments and made assumptions for modeling based solely on the presence or absence of bioturbation. Many other studies have considered the effects of many other variables but the way the abstract and text is written, a reader could get the incorrect impression that the

assumptions in the Boyle paper reflect a general consensus about P cycling. In my opinion, it would be preferable to focus on the direct results of this nice study in the abstract and save the consideration of the Boyle assumptions along with other ideas on factors affecting P cycling in sediments for the introduction and discussion.

My main criticism of this manuscript is that the concluding sentences of the abstract are too broad. There are many caveats in extrapolating the results of this study to marine systems in general. The high sediment accumulation rates and organic matter composition set this environment apart from normal marine systems. Fortunately, these caveats, as well as other potential concerns, are discussed in the last paragraph of the manuscript. These caveats should be reflected in the abstract.
* * *

---

## Referee Comment (RC2) · Peter Kraal (Referee) · 9 Dec 2020

Review of bg-2020-340

With interest I have read this paper, in which the authors explore the impact of bioturbation on OM and P burial in salt marsh sediments. The findings are presented within the context of the rise of burrowing animals and the biogeochemical impact of burrowing on the ocean-climate system. The paper is well-written (I have almost no technical comments on spelling/grammar) and presents an interesting, well-constrained test case to evaluate assumptions regarding causal relationships between ecology and biogeochemistry, which are also important to parametrize regional and global biogeochemical models.

[Figure]

The manuscript suffers from a rather limited exploration of the data, shackled by a strict context of two "hypotheses" which are presented as key foundations of our understanding of (the evolution of) P cycling, while a broader context and more in-depth analysis of the presented data and the processes that are discussed would be beneficial. The focus on the validation of ancient (open) ocean scenarios with results from terrestrially-impacted salt marsh sediments is a bit of a stretch to me.

I have a few considerations for the authors, with which I hope to provide fair and constructive feedback to improve the manuscript.

Bioturbation| To my understanding, bioturbation has a rather complex impact on surface sediments bringing oxidants into reducing deeper sediment and the other way around, explored in some detail by modellers (e.g. Boudreau and others) and experimenters (Aller and others). The authors start the paper by only in very general terms mentioning that bioturbation impacts biogeochemical processes without a mechanistic backdrop, and I do not see further process-based explanation later in the introduction. The 'burrowing revolution' and 'bioturbation' are not explicitly linked in the text, so bioturbation is not really defined. Starting like this, the paper is only accessible to people already intimately familiar with processes and effects of bioturbation. I think the authors move to fast here and should reconsider how to establish the general context of their study, and in particular inform the reader at a mechanistic level what it is that bioturbation does to the sediment to impact the "geochemical cycling and burial of elements".

Ordering of information | The manuscript, especially the introduction, seems to have a style in which cause-effect relationships are mentioned, but the underlying mechanisms are only mentioned later. For instance, how do C and P burial control atmospheric $O_2$ and $CO_2$ (p2. L3-4), primary productivity should probably be mentioned here already, instead of later (L11-14). And how does bioturbation decrease the (C/P)org ratio of buried organic material (p2. L11-16, L22-23)? I find the phrasing somewhat confusing, because from the context (also from the refs) I gather that bioturbation contributes additional organic P from diagenesis of polyphosphates, which

I not apparent from the text "The key assumption is that bioturbation decreases the carbon-to-phosphorus ratio of the organic matter (Corg : Porg) that is buried in marine sediments". The authors should critically assess whether they provide the required mechanistic backdrop (in a timely manner) in these and other cases. The reader should not be left with questions of why lingering in their mind, only to find answers later on (or not at all).

The hypotheses on P burial and bioturbation | The manuscript sets out to test two supposed key assumptions on the links between P burial and bioturbation: (1) bioturbation boosts Porg burial (initially polyphosphates) and thus results in lower overall (C/P)org and (ii) inorganic P burial is not affected by bioturbation. The salt marshes offer a unique system where bottom waters are oxygenated and thus the isolated effect of bioturbation can be studied, rather than having to untangle anoxia and burrowing intensity. In setting this scene of key assumptions, the authors lean pretty heavily on one publication, Boyle et al., 2014. In this publication, a highly simplified P cycle is used in a general model with a simplistic mathematical representation of the effect of bioturbation, namely by making the total organic P burial flux dependent on a bioturbation factor that apportions P burial between (prescribed) high-C/P (laminated) and low-C/P (bioturbated) sediment. I find the current balance of the manuscript, where a local P study in salt marshes is used mostly to challenge some broad assumptions in a global modelling study not very strong. Added to this, the current manuscript incorrectly implies that the Boyle "assumptions" represent our collective understanding of P burial and bioturbation. I have noticed that the other reviewer was also struck by this, and I agree that the findings can be much better placed within the broader context of factors controlling P burial in sediments. Furthermore, I wonder about the representation of the "assumptions": the authors mention that Boyle et al. assume that bioturbation has no effect on Pinorg burial. However, the model adaptation by Boyle et al. includes redox-driven FeOx - P coupling that shows a strong increase in sedimentary FeOx-P content upon bioturbation (and subsequent counter-intuitive effects on ocean oxygenation). This suggests that bioturbation boosts Porg formation and burial but also FeOx-P

(Pinorg) burial? This study complements that understanding nicely by showing that this effect seems short-lived. All in all, the study would profit from a shift away from re-butting "assumptions" in a simplistic P cycling model, and the authors should carefully evaluate their representation of the "assumptions" in Boyle's et al. 2014.

Geochemical context | Wording on p8. L3 that "Phosphorus adsorbs onto iron oxide minerals, which are introduced at depth by the downward mixing of benthic fauna" is slightly ambiguous as it could mean P-enriched FeOx being mixed into the sediment, or P adsorbing onto FeOx that were mixed deeper into the sediment. Perhaps seems trivial, but good to clarify. This brings me to a more general point: identification of geo-chemical processes (at depth). The authors mention reductive dissolution, release of P, precipitation of Fe-P, but all without showing any data for pertinent dissolved species such as Ca, Fe and P. As such, the actual (redox) P cycle remains speculative, while the availability of these cores from ponds offers the possibility to address a lot of these questions by measuring pore-water chemistry. The authors refer to data in Van der Velde et al. 2020 (e.g. dissolved Fe), it seems that the detailed geochemical function-ing of these sediments was evaluated in that paper and a small side study on P was taken from the dataset for this paper. As a result, the current manuscript is very light on geochemical context. This does not only apply to the pore-water, but also the sediment chemistry: the authors mention limited Ca-P formation in the cores and contrast this with previous studies (p9. L4-16). Solid-phase chemistry (e.g. sediment composition) could further support statements on the likely fate of P released from organic material.

Comparing across timescales and depositional systems | The authors compare C/P ratios of salt marsh sediment that, according to the text, probably receive mostly terres-trial organic matter with C/P ratios of ancient sediments undergoing (tens to hundreds of) millions of years of diagenesis. Having studied such ancient sediments myself, I doubt how informative the (C/P)org ratios in such materials really are (e.g. Kraal et al., BG, 2012), which is in fact the reason why Corg/Preactive ratios are commonly used for ancient sediments. The transformation of Porg or PFe to stable authigenic

phases over geological timescales is very well established for a range of marine settings. A complete lack of such signals in the current data (at least on the very short timescales considered) may say something rather specific for the type of sediment under consideration. Similarly, polyphosphates are becoming recognized more and more as an important (intermediate) P sink in many marine systems, and in fact are the key intermediate for the bioturbation-Porg burial link considered here, but seem to play no role whatsoever in the studied sediment? For me, this truly raises questions about the validity of comparing across such disparate timescales and depositional conditions. Currently, the authors deal with this by making bold claims in the abstract and the discussion, followed by listing all kinds of caveats. The central role of polyphosphates in booting Porg burial is mentioned in the into but not really discussed in much detail at all. This does not make the manuscript particularly strong. I think it would be better to discuss the data for what they are, an interesting look into P cycling (assuming supporting data such as dissolved Fe and P will be added) in salt marsh sediments which offer a chance at investigating the role of bioturbation there. A section of the discussion could then be spent on how this may impact our more general understanding of the impact of bioturbation on P burial, but the system is too specific and the dataset too limited to support the current setup.

P profiles | The authors present the conclusions from the POP data and (C/P)org ratios with no caution in the abstract but with great caution in the discussion: the organic matter is probably mostly terrestrial (C/P > 500 under oxic conditions) and may behave differently than "normal" (algal) organic material in marine sediments. There is no characterization of the OM. This puts quite some strain on the validity of linking the salt marsh results to global ocean (modelling) studies. Having said that, the POP profiles shown in Fig. 2 provide more insight. The non-bioturbated POP profile shows a steady decrease ($\sim$ 50%) with depth that follows TOC, which suggests that OM is being degraded at appreciable rates (can this be ascertained for instance from NH4+ profiles (Van der Velde, 2020), or are other processes responsible for the down-core decrease?). If the authors can evaluate OM degradation rates from the profiles and

show that these are comparable to down-core TOC and POP profiles in more marine settings, the results presented here regain validity as testing ground for ocean processes. In short, I would recommend to: (i) include the disclaimer about different types of organic matter in appropriate places such as the abstract and (ii) dig a little deeper into the salt-marsh POP profiles and their relation to more generic marine settings. A second point is the P budget. The authors present Porg and Pinorg in different figures and do not show the total P profiles; this makes is a bit difficult to assess down-core trends in P, it would be nice to see all P profiles and dissolved P) together, followed by the boxplots that assess the differences between sites. I am also curious to know whether these sediments are a net source or sink of P based on diffusive exchange between the bottom water and sediment.

Specific p6. L10. non-bioturbated sediments have strongest gradients in Fig. 2 p8. L7-17. Discussion starts to infiltrate the Results section at this point. Also creating unnecessary overlap with first section of Discussion.

---

## Author Comment (AC1) · 15 Dec 2020

We thank the reviewers for the constructive comments given on the manuscript.

Please find in the supplement our responses to the specific comments made by the reviewers

Please also note the supplement to this comment:
https://bg.copernicus.org/preprints/bg-2020-340/bg-2020-340-AC1-supplement.pdf

---

## Author Response (AR1)

Title: "Bioturbation has a limited effect on phosphorus burial in salt marsh sediments"
Tracking #: bg-2020-340
Authors: van de Velde et al.

Referee #1:

This is a well presented study that compares P burial and preservation in salt marsh ponds under oxygenated vs anoxic conditions. A nice aspect of this study is that major variables that are known to affect P deposition, such as sediment accumulation rate and organic matter composition, are very similar in the different ponds. Thus, the effect of other variables such as oxygen and bioturbation can be isolated. The paper is written in such a way that it appears a primary motivation for this work is to refute statements made in the paper by Boyle et al., 2014. From reading this paper it sounds like the Boyle paper did not consider many variables that can affect P cycling in sediments and made assumptions for modeling based solely on the presence or absence of bioturbation. Many other studies have considered the effects of many other variables but the way the abstract and text is written, a reader could get the incorrect impression that the assumptions in the Boyle paper reflect a general consensus about P cycling. In my opinion, it would be preferable to focus on the direct results of this nice study in the abstract and save the consideration of the Boyle assumptions along with other ideas on factors affecting P cycling in sediments for the introduction and discussion.

My main criticism of this manuscript is that the concluding sentences of the abstract are too broad. There are many caveats in extrapolating the results of this study to marine systems in general. The high sediment accumulation rates and organic matter composition set this environment apart from normal marine systems. Fortunately, these caveats, as well as other potential concerns, are discussed in the last paragraph of the manuscript. These caveats should be reflected in the abstract.

We are thankful for the positive evaluation of our manuscript. The way we present the paper seems to create a misconception of our intent, as our primary motivation was not to refute the assumptions made in Boyle et al. (2014) (and later on by some of the authors of this manuscript in van de Velde et al., 2018), but rather to find direct observational support for these assumptions. Because of the confusion our presentation created for both referees, we restructured our abstract and introduction, and refer to the potential implications for the global impact of the evolution of bioturbation in the discussion section.

We additionally rewrote the abstract to reflect the content of our manuscript better (including the caveats). "*It has been hypothesised that the evolution of animals during the Ediacaran-Cambrian transition stimulated the burial of phosphorus in marine sediments. This assumption is centrally based on data compilations from marine sediments deposited under oxic and anoxic bottom waters. Since anoxia excludes the presence of infauna and sediment reworking, the observed differences in P burial are assumed to be driven by the presence of bioturbators. This reasoning however ignores the potentially confounding impact of bottom water oxygenation on phosphorous burial.*
*Here, our goal is to test the idea that bioturbation increases the burial of organic phosphorus, while accounting for bottom water oxygenation. We present solid-phase phosphorus speciation data from salt marsh ponds with and without bioturbation (Blakeney salt marsh, Norfolk, UK).*

*In both cases, the pond sediments are exposed to oxygenated bottom waters and so the only difference is the presence/absence of bioturbating macrofauna. Our data reveal that the rate of Porg and of Pinorg burial are indistinguishable between bioturbated and non-bioturbated sediments. A large terrestrial fraction of organic matter and higher sedimentation velocity than generally found in marine sediments ($0.3 \pm 0.1$ cm yr-1) may partially impact these results. However the absence of a clear effect of bioturbation on total P burial puts into question the presumed importance of bioturbation for phosphorus burial."*

Referee #2:

With interest I have read this paper, in which the authors explore the impact of bioturbation on OM and P burial in salt marsh sediments. The findings are presented within the context of the rise of burrowing animals and the biogeochemical impact of burrowing on the ocean-climate system. The paper is well-written (I have almost no technical comments on spelling/grammar) and presents an interesting, well-constrained test case to evaluate assumptions regarding causal relationships between ecology and biogeochemistry, which are also important to parametrize regional and global biogeochemical models.

The manuscript suffers from a rather limited exploration of the data, shackled by a strict context of two "hypotheses" which are presented as key foundations of our understanding of (the evolution of) P cycling, while a broader context and more in-depth analysis of the presented data and the processes that are discussed would be beneficial. The focus on the validation of ancient (open) ocean scenarios with results from terrestrially-impacted salt marsh sediments is a bit of a stretch to me.

I have a few considerations for the authors, with which I hope to provide fair and constructive feedback to improve the manuscript.

We appreciate the interest and the in-depth and constructive review effort. We agree with your main concern (as also mentioned by referee #1), and have rewritten the introduction to shift the focus away from the assumptions of the global biogeochemical models of Boyle et al. (2014) and van de Velde et al. (2018). We also introduced more caution into our abstract, added more mechanistic context in the introduction and included more justification as to why we believe our field site is representative to more open marine settings.

Bioturbation| To my understanding, bioturbation has a rather complex impact on surface sediments bringing oxidants into reducing deeper sediment and the other way around, explored in some detail by modellers (e.g. Boudreau and others) and experimenters (Aller and others). The authors start the paper by only in very general terms mentioning that bioturbation impacts biogeochemical processes without a mechanistic backdrop, and I do not see further process-based explanation later in the introduction. The 'burrowing revolution' and 'bioturbation' are not explicitly linked in the text, so bioturbation is not really defined. Starting like this, the paper is only accessible to people already intimately familiar with processes and effects of bioturbation. I think the authors move to fast here and should reconsider how to establish the general context of their study, and in particular inform the reader at a mechanistic level what it is that bioturbation does to the sediment to impact the "geochemical cycling and burial of elements".

This is a valid point, and we have rewritten the introduction to provide an explicit link between 'burrowing revolution' and 'bioturbation' and also provided more mechanistic context.

The opening paragraph of the introduction has been rewritten as:

'*The evolution of animals near the Ediacaran-Cambrian transition (~542 million years ago - Ma) was a major evolutionary event in Earth's history (Mángano and Buatois, 2017; Meysman et al., 2006; Wood et al., 2019). Early benthic animals developed the ability to burrow (the so-called 'burrowing revolution'; Meysman et al., 2006), which profoundly changed the geochemical cycling and burial of elements in the seafloor (Aller, 1977; McIlroy and Logan,*

*1999; Meysman et al., 2006). Benthic fauna affect the seafloor in two separate ways; by reworking of solid-phase particles (bio-mixing) and by flushing of burrows (bio-irrigation), lumped together under the term 'bioturbation' (Kristensen et al., 2012). Bio-mixing and bio-irrigation can have distinct effects on organic carbon mineralisation and early diagenesis (Kostka et al., 2002; van de Velde and Meysman, 2016). For instance, bio-irrigation can promote aerobic respiration by flushing oxygenated bottom waters into deeper anoxic horizons (Archer and Devol, 1992; van de Velde and Meysman, 2016), whereas bio-mixing transports fresh organic from the sediment-water interface into the anoxic zone, thus stimulating anaerobic mineralisation pathways (Berner and Westrich, 1985; van de Velde and Meysman, 2016).'*

We have also extended the second paragraph of the introduction to include a more general description of the potential role of bioturbation in regulating P burial;

*'Bioturbation has been proposed to play a key role in the sedimentary phosphorus (P) cycle. Via bio-mixing, bioturbating organisms transport P that is adsorbed on iron oxides from the oxic zone at the sediment surface into deeper sedimentary layers (Slomp et al., 1996). In the anoxic zone of the sediment, these iron oxides are reduced, and P is released further away from the sediment-water interface (SWI). Accordingly, P is retained longer in the sediment, which could stimulate the precipitation and eventual burial of inorganic P (Pinorg) minerals, such as apatite (Slomp et al., 1996). Via bio-irrigation, benthic fauna can increase the availability of oxygen in the sediment (Volkenborn et al., 2019), which could stimulate the production of microbial polyphosphate compounds within the sediment column (Dale et al., 2016). Microbial polyphosphates are generated during the breakdown of organic matter under oxic conditions (Diaz et al., 2008), and can be converted into more refractory organic P (Porg; as for example phosphate esters or phosphonates) or inorganic P minerals during diagenesis, which would then constitute a permanent Porg burial sink (Berner et al., 1993; Van Cappellen and Ingall, 1994; Diaz et al., 2008; Goldhammer et al., 2010; Ingall and Jahnke, 1997). Overall, bioturbation could increase the burial of organic and inorganic P in the sediment. Accordingly, it has been hypothesised that the rise of bioturbation at the Ediacaran-Cambrian boundary increased the burial of P in marine sediments (Boyle et al., 2014; Dale et al., 2016; van de Velde et al., 2018). Marine P burial plays a key role in the long-term functioning of the Earth system, because P is considered the long-term limiting nutrient for marine primary productivity (Van Cappellen and Ingall, 1996). If more P becomes buried, photosynthesis would decrease, thereby limiting the overall burial of Corg in the seafloor, subsequently increasing atmospheric CO2 and lowering atmospheric O2 (Bergman et al., 2004; Berner, 1982). Hence, the rise of bioturbation may have increased atmospheric CO2 concentrations and decreased atmospheric O2 concentrations, thus inducing warmer climatic conditions and more widespread ocean anoxia (Boyle et al., 2014; van de Velde et al., 2018).'*

Ordering of information | The manuscript, especially the introduction, seems to have a style in which cause-effect relationships are mentioned, but the underlying mechanisms are only mentioned later. For instance, how do C and P burial control atmospheric O2 and CO2 (p2. L3-4), primary productivity should probably be mentioned here already, instead of later (L11-14). And how does bioturbation decrease the (C/P)org ratio of buried organic material (p2. L11-16, L22-23)? I find the phrasing somewhat confusing, because from the context (also from the refs)

I gather that bioturbation contributes additional organic P from diagenesis of polyphosphates, which I not apparent from the text "The key assumption is that bioturbation decreases the carbon-to-phosphorus ratio of the organic matter (Corg : Porg) that is buried in marine sediments". The authors should critically assess whether they provide the required mechanistic backdrop (in a timely manner) in these and other cases. The reader should not be left with questions of why lingering in their mind, only to find answers later on (or not at all).

We have rewritten the introduction (see previous response).

The hypotheses on P burial and bioturbation | The manuscript sets out to test two supposed key assumptions on the links between P burial and bioturbation: (1) bioturbation boosts Porg burial (initially polyphosphates) and thus results in lower overall (C/P)org and (ii) inorganic P burial is not affected by bioturbation. The salt marshes offer a unique system where bottom waters are oxygenated and thus the isolated effect of bioturbation can be studied, rather than having to untangle anoxia and burrowing intensity. In setting this scene of key assumptions, the authors lean pretty heavily on one publication, Boyle et al., 2014. In this publication, a highly simplified P cycle is used in a general model with a simplistic mathematical representation of the effect of bioturbation, namely by making the total organic P burial flux dependent on a bioturbation factor that apportions P burial between (prescribed) high-C/P (laminated) and low-C/P (bioturbated) sediment. I find the current balance of the manuscript, where a local P study in salt marshes is used mostly to challenge some broad assumptions in a global modelling study not very strong. Added to this, the current manuscript incorrectly implies that the Boyle "assumptions" represent our collective understanding of P burial and bioturbation. I have noticed that the other reviewer was also struck by this, and I agree that the findings can be much better placed within the broader context of factors controlling P burial in sediments. Furthermore, I wonder about the representation of the "assumptions": the authors mention that Boyle et al. assume that bioturbation has no effect on Pinorg burial. However, the model adaptation by Boyle et al. includes redox-driven FeOx - P coupling that shows a strong increase in sedimentary FeOx-P content upon bioturbation (and subsequent counter-intuitive effects on ocean oxygenation). This suggests that bioturbation boosts Porg formation and burial but also FeOx-P (Pinorg) burial? This study complements that understanding nicely by showing that this effect seems short-lived. All in all, the study would profit from a shift away from rebutting "assumptions" in a simplistic P cycling model, and the authors should carefully evaluate their representation of the "assumptions" in Boyle's et al. 2014.

We agree that the goals of this study do not come across as we had intended. Therefore, we have adapted the introduction, to avoid leaning to heavily on the assumptions made in the model studies of Boyle et al., 2014 and van de Velde et al., 2018. Instead we frame the study as an ideal testcase to assess the impact of bioturbation on P burial. The discussion of the two assumptions has been removed altogether, and we provide more mechanistic discussion to our observations.

Geochemical context | Wording on p8. L3 that "Phosphorus adsorbs onto iron oxide minerals, which are introduced at depth by the downward mixing of benthic fauna" is slightly ambiguous as it could mean P-enriched FeOx being mixed into the sediment, or P adsorbing onto FeOx that were mixed deeper into the sediment. Perhaps seems trivial, but good to clarify.

Rewritten as '*Phosphorus adsorbed onto iron oxide minerals is transported from the sediment-water interface to the deeper sedimentary layers by the downward mixing of benthic fauna*'

This brings me to a more general point: identification of geochemical processes (at depth). The authors mention reductive dissolution, release of P, precipitation of Fe-P, but all without showing any data for pertinent dissolved species such as Ca, Fe and P. As such, the actual (redox) P cycle remains speculative, while the availability of these cores from ponds offers the possibility to address a lot of these questions by measuring pore-water chemistry. The authors refer to data in Van der Velde et al. 2020 (e.g. dissolved Fe), it seems that the detailed geochemical functioning of these sediments was evaluated in that paper and a small side study on P was taken from the dataset for this paper. As a result, the current manuscript is very light on geochemical context. This does not only apply to the pore-water, but also the sediment chemistry: the authors mention limited Ca-P formation in the cores and contrast this with previous studies (p9. L4-16). Solid-phase chemistry (e.g. sediment composition) could further support statements on the likely fate of P released from organic material.

This is not entirely correct. The van de Velde et al. GCA paper that is referred to has a different scope, as there we tried to identify the cause of the interesting sediment redox dichotomy found in these salt marshes. The scope of this study is entirely different, as we set out to quantify the burial of phosphorus in bioturbated and unbioturbated conditions. You are correct that more detailed pore water analysis would help to shed more light on the redox processes, but with the limited resources (time, people …) we had to make a choice between analysing fewer cores in detail, or purely looking at the phosphorus minerals in more cores. We chose the latter, as natural variability makes it difficult to distinguish differences between the different sediment types.

We agree that extensive pore water data could help to elucidate (to some extent) the geochemical processes within the sediment. However, we do not believe that pore water data are required for our ultimate goal: assessing P burial in bioturbated and non-bioturbated conditions. We feel that the solid-phase data is strong enough to answer that question. For example, we mention limited Ca-P formation because we were unable to detect the operationally defined 'authigenic P' fraction. We have analysed the solid-phase phosphorus speciation for all samplings (and the more detailed SEDEX extraction for 2015 and 2016), we are unsure how sediment composition or pore-water data would have altered our conclusion with respect to the P-burial phases.

Comparing across timescales and depositional systems | The authors compare C/P ratios of salt marsh sediment that, according to the text, probably receive mostly terrestrial organic matter with C/P ratios of ancient sediments undergoing (tens to hundreds of) millions of years of diagenesis. Having studied such ancient sediments myself, I doubt how informative the (C/P)org ratios in such materials really are (e.g. Kraal et al., BG, 2012), which is in fact the reason why Corg/Preactive ratios are commonly used for ancient sediments. The transformation of Porg or PFe to stable authigenic phases over geological timescales is very well established for a range of marine settings. A complete lack of such signals in the current data (at least on the very short timescales considered) may say something rather specific for the type of sediment under consideration.

As you say, transformation of Porg/PFe to stable authigenic phases occur over geological timescales, whereas our cores represent ~60 years. As we elaborate in the discussion, if there is no difference between the bioturbated and unbioturbated sites below the bioturbated zone, is

there any reason to assume that bioturbation has left an imprint on the sediment that only later becomes apparent?

*We agree that Corg/Preactive profiles would be more informative. Unfortunately, the extraction method used in 2018 does not allow the differentiation between the different Pinorg phases, and we can thus not account for the detrital P phases, which would introduce a bias in the Corg/Preactive ratios. Instead, we have rewritten the results and discussion sections and just use the (C/P)org ratios as an indicator of the type of organic matter.*

Similarly, polyphosphates are becoming recognized more and more as an important (intermediate) P sink in many marine systems, and in fact are the key intermediate for the bioturbation-Porg burial link considered here, but seem to play no role whatsoever in the studied sediment? For me, this truly raises questions about the validity of comparing across such disparate timescales and depositional conditions. Currently, the authors deal with this by making bold claims in the abstract and the discussion, followed by listing all kinds of caveats. The central role of polyphosphates in booting Porg burial is mentioned in the into but not really discussed in much detail at all. This does not make the manuscript particularly strong. I think it would be better to discuss the data for what they are, an interesting look into P cycling (assuming supporting data such as dissolved Fe and P will be added) in salt marsh sediments which offer a chance at investigating the role of bioturbation there. A section of the discussion could then be spent on how this may impact our more general understanding of the impact of bioturbation on P burial, but the system is too specific and the dataset too limited to support the current setup.

*To our knowledge, the bioturbation-Porg link has only been demonstrated theoretically in a diagenetic model study of Dale et al. (2016 cited in the main text). These authors also only considered polyphosphate formation under oxic conditions, as this has been shown to occur in the water column by Diaz et al. (2008, cited in the main text). In another study however, it was shown that polyphosphates are also formed under anoxic conditions (Goldhammer et al., 2010). We are unaware of any studies showing conclusive evidence for the formation of polyphosphates in sediments (the fact that they show up in surface sediments is likely due to settling from the water column). Our data also does not allow us to say whether or not they play a role, as we cannot differentiate between bulk organic P and microbial polyphosphates. As mentioned above, our goal was to look at the role of bioturbation for the burial of solid phase P. We do agree that future studies could look into more detail at the actual diagenesis of P, including the potential role of polyphosphates.*

*We have now expanded the discussion of the potential importance of polyphosphates: "Bioturbation is believed to impact Porg burial by stimulating the formation of polyphosphates, which can act as an intermediate for the formation of apatite or more refractory organic P compounds (Berner et al., 1993; Goldhammer et al., 2010). Polyphosphates are formed during aerobic respiration of organic matter (Diaz et al., 2008), and because bio-irrigation stimulates aerobic respiration in the sediments (Archer and Devol, 1992; van de Velde and Meysman, 2016), it has been hypothesised that more polyphosphates could be formed in bioturbated sediments (Dale et al., 2016). However, other studies have shown that polyphosphates and the subsequent conversion to apatite also occurs under anoxic conditions (Goldhammer et al., 2010), so it is questionable whether periodic influshing of oxygenated water would have a large impact on polyphosphate formation. Unfortunately, our analysis method does not allow us to*

*differentiate between bulk organic P and microbial polyphosphates to investigate whether polyphosphate formation plays a role in both bioturbated and non-bioturbated sediments. This would be an interesting avenue for future research.*"

We already discussed the caveats about upscaling our site to the global ocean, and only in the last paragraph now briefly mention the potential implications of our results for our understanding of the impact of the evolution of bioturbation on the Earth System.

P profiles | The authors present the conclusions from the POP data and (C/P)org ratios with no caution in the abstract but with great caution in the discussion: the organic matter is probably mostly terrestrial (C/P > 500 under oxic conditions) and may behave differently than "normal" (algal) organic material in marine sediments. There is no characterization of the OM. This puts quite some strain on the validity of linking the salt marsh results to global ocean (modelling) studies. Having said that, the POP profiles shown in Fig. 2 provide more insight. The non-bioturbated POP profile shows a steady decrease (~ 50%) with depth that follows TOC, which suggests that OM is being degraded at appreciable rates (can this be ascertained for instance from $NH_4^+$ profiles (Van der Velde, 2020), or are other processes responsible for the down-core decrease?). If the authors can evaluate OM degradation rates from the profiles and show that these are comparable to down-core TOC and POP profiles in more marine settings, the results presented here regain validity as testing ground for ocean processes.

There has been some characterization of the OM, using the C:N:P ratios found in the sediment (Fig. 2) and of the surrounding vegetation (Suppl. Table 3), and as mentioned, we have included that in the discussion. We now include an extra discussion to explain why this field site has some relevance for marine sedimentary processes (for example by showing that the mineralization rates are in the same range as found for marine sediment); '*Nevertheless, the POC profiles suggest an appreciable amount of organic matter is being degraded in the salt marsh pond sediments (Figure 2). Indeed, organic matter mineralisation rates based on POC and nutrient profiles range from 6-38 mmol C m-2 d-1, which is comparable to rates observed in shallow marine sediments (Burdige, 2007) and thus indicates that the Blakeney salt marsh site shows a comparable metabolic activity to marine sediments. The observation then still remains; bioturbation seems to have no significant impact on Porg burial.*'

We also added more caution in our abstract (see reply to Referee 1).

In short, I would recommend to: (i) include the disclaimer about different types of organic matter in appropriate places such as the abstract and (ii) dig a little deeper into the salt-marsh POP profiles and their relation to more generic marine settings. A second point is the P budget. The authors present Porg and Pinorg in different figures and do not show the total P profiles; this makes is a bit difficult to assess down-core trends in P, it would be nice to see all P profiles and dissolved P) together, followed by the boxplots that assess the differences between sites. I am also curious to know whether these sediments are a net source or sink of P based on diffusive exchange between the bottom water and sediment.

We have amended the abstract to more truthfully represent the contents of the manuscript, and added some more discussion on the mineralization of organic matter in our field site. We also show total P alongside particulate organic and inorganic phosphorus in Fig. 2.

Specific
p6. L10. non-bioturbated sediments have strongest gradients in Fig. 2

Changed

p8. L7-17. Discussion starts to infiltrate the Results section at this point. Also creating unnecessary overlap with first section of Discussion.

We changed that section to: "*The increased inventory of Pexch and PFe at depth seemingly does not lead to more precipitation of Pauth (Suppl. Table 5), since we do not find detectable formation of authigenic apatite, the concentrations of which are negligible throughout all sediment cores in both ponds (< 1 µmol g-1). While Pinorg concentrations are high in the top layers, they strongly decrease with depth, thus Pinorg is not efficiently buried. Overall, we find that the burial rate of Pinorg is 38 ± 3 µmol m-2 d-1 in bioturbated sediments, and 36 ± 3 µmol m-2 d-1 in the non-bioturbated sediment, and this difference is not significant (p > 0.1; Suppl. Table 4).*"

And we moved the part about the potential vivianite formation to the discussion.